# Atg18 oligomer organization in assembled tubes and on lipid membrane scaffolds

Daniel Mann [1,2,8], Simon A. Fromm [3,4,8], Antonio Martinez-Sanchez [5], Navin Gopaldass[6], Ramona Choy[1,2], Andreas Mayer [6] & Carsten Sachse [1,2,7] ✉

Autophagy-related protein 18 (Atg18) participates in the elongation of early autophagosomal structures in concert with Atg2 and Atg9 complexes. How Atg18 contributes to the structural coordination of Atg2 and Atg9 at the isolation membrane remains to be understood. Here, we determined the cryo-EM structures of Atg18 organized in helical tubes, Atg18 oligomers in solution as well as on lipid membrane scaffolds. The helical assembly is composed of Atg18 tetramers forming a lozenge cylindrical lattice with remarkable structural similarity to the COPII outer coat. When reconstituted with lipid membranes, using subtomogram averaging we determined tilted Atg18 dimer structures bridging two juxtaposed lipid membranes spaced apart by 80 Å. Moreover, lipid reconstitution experiments further delineate the contributions of Atg18's FRRG motif and the amphipathic helical extension in membrane interaction. The observed structural plasticity of Atg18's oligomeric organization and membrane binding properties provide a molecular framework for the positioning of downstream components of the autophagy machinery.

Macroautophagy (from here on referred to as autophagy) degrades long-lived proteins, macromolecular complexes, and organelles in order to recuperate the molecular building blocks for the cell[1–5]. In this function, autophagy is critical for cellular maintenance and homeostasis in eukaryotes, and dysregulation is implicated in neurodegeneration, cancer, inflammation, and infection[6,7]. During autophagy a double-membrane organelle called autophagosome is formed de novo[8], engulfs cytosolic contents, and fuses with the lysosome[9]. In the meantime, more than 40 gene products have been elucidated, which are organized in six major functional protein complexes: Atg1 kinase[10], class III phosphatidylinositol-3-kinase (PI-3K)[11], Atg9 lipid scramblase[12], the Atg2-Atg18 lipid transfer complex[13], and two ubiquitin-like conjugation systems[14].

In budding yeast *Saccharomyces cerevisiae*, the Atg1 complex initiates the pre-autophagosomal structure (PAS) by phase separation[15]. Subsequently, small lipid vesicles containing trimeric Atg9 are delivered to the early isolation membrane (IM)[16]. Initially, the IM was observed in direct contact with membranes of the endoplasmic reticulum (ER)[17,18]. Moreover, COPII vesicles and their molecular components were also shown to be involved in lipid provision to the autophagosome when the IM is in contact with the ER exit site[19,20]. While the Atg2-Atg18 complex was found to tether the pre-autophagosomal membrane to the ER[21] including Atg9[22], at later stages the Atg2-Atg18 complex was observed localized to the expanding tips of the IM[23,24]. A recent Atg9 cryo-EM structure revealed the trimeric organization and biochemical activity as a lipid scramblase[25,26]. Atg2 is a very large 1592 aa rod-shaped protein that was shown to function as an inter-membrane lipid transfer protein[27–29] and has homology to a recently resolved VPS13 structure revealing the principal architecture of a lipid slide[30].

[1]Ernst-Ruska Centre 3/Structural Biology, Forschungszentrum Jülich, Wilhelm-Johnen-Straße, Jülich, Germany. [2]Institute for Biological Information Processing 6/Structural Cellular Biology, Forschungszentrum Jülich, Wilhelm-Johnen-Straße, Jülich, Germany. [3]Structural and Computational Biology Unit, European Molecular Biology Laboratory (EMBL), Heidelberg, Germany. [4]EMBL Imaging Centre, European Molecular Biology Laboratory, Heidelberg, Germany. [5]Department of Information and Communications Engineering, Faculty of Computers Sciences, University of Murcia, Murcia, Spain. [6]Department of Biochemistry, University of Lausanne, Epalinges, Switzerland. [7]Department of Biology, Heinrich Heine University, Universitätsstr. 1, Düsseldorf, Germany. [8]These authors contributed equally: Daniel Mann, Simon A. Fromm. ✉e-mail: c.sachse@fz-juelich.de

Closely linked to Atg2 is Atg18 that early on was shown to be essential for the progression of autophagy[31,32]. Atg18 belongs to the protein family of β-propellers that bind polyphosphoinositides (PROPPIN). The corresponding primary structure contains a series of conserved WD40 repeats forming the 6 or 7 blades of a β-propeller fold. Initially, the X-ray crystal structures of Hsv2, which is a close relative of the PROPPIN family[33,34], and the *Pichia angusta* homolog of Atg18 had been determined[35]. More recently, the structure of *S. cerevisiae* Atg18 was elucidated in the presence of phosphate and citrate, revealing binding sites for phosphatidylinositol-3-phophate (PI3P) and phosphatidylinositol-3,5-bisphophate PI(3,5)P$_2$ (PDB-IDs 5LTD, 5LTG)[36]. Mutations in conserved binding sites for PI3P and PI(3,5)P$_2$ also known as the FRRG motif affect the progression of the cytoplasm-to-vacuole targeting (Cvt) pathway and autophagy as well as vacuolar morphology[37,38] supporting the specific membrane adapter role of Atg18. Moreover, Atg18 contains three up to 120 aa loop regions (i.e., 6AB, 6CD, and 7AB) that possess a high content of predicted structural disorder and emanate from the blades of the β-propeller scaffold. The 6CD loop (324–406) of Atg18, for instance, participates in membrane binding[35,39] and a subsegment was proposed to form an amphipathic helix in the membrane environment[40]. Moreover, lipid reconstitution experiments of Atg18 with giant unilamellar vesicles were shown to remodel and tubulate membranes thus mediating membrane scission[40]. Once bound to membranes, Atg18 was found to undergo oligomer formation as observed by cross-linking mass spectrometry[35]. In addition to residues P72/R73[41], the 7AB loop (430–460) of Atg18 was shown to bind to the rod-shaped Atg2 molecule[36] that is thought to bridge the PI3P-rich IM on one end with a membrane lacking PI3P like the ER at the opposite end[27].

In addition to Atg18, Atg21 and Hsv2 are structurally related yeast PROPPIN family members involved in membrane trafficking[42]. It was found that Atg18 is essential for autophagy and the related Cvt pathway whereas Atg21 is solely for Cvt[13,37]. The specific PI3P binding and downstream Atg recruitment function of Atg18 and Atg21, however, can be compensated by one another[43]. Hsv2 was shown to be required for microautophagy of the nucleus[44] and was the first of the PROPPINs to be structurally resolved[33,34]. In higher eukaryotes, Atg18 family proteins are also known as WD-repeat proteins interacting with phosphoinositides (PIP) (WIPI) with a total of four isoforms occurring in human cell lines. WIPI1, WIPI2, and WIPI4 have been shown to be directly linked to autophagic progression[45]. Overexpression of WIPI1 led to large and elongated light microscopic punctae in human cell lines[46]. WIPI2 recruits the ATG12-ATG5-ATG16L1 LC3 conjugation machinery[47] and WIPI4 was demonstrated to biochemically interact with ATG2A and assist in the lipid transfer activity[28]. The Atg18/WIPI family constitutes a complementary network of proteins involved in autophagy and autophagy-related cellular processes.

Although various aspects of Atg18 contributions to the autophagy pathway have been investigated, the involvement and structural role of oligomeric species at the IM remains poorly understood. In this study, we applied electron cryo-microscopy (cryo-EM) to solve structures of helically assembled, soluble, and membrane-bound Atg18 using high-resolution cryo-EM structure determination as well as subtomogram averaging. Interestingly, the studied Atg18 helical assembly has striking architectural similarities to the COPII coat. The observed structural plasticity of Atg18 indicates that multiple states of monomeric, dimeric, and oligomeric assemblies need to be considered for interacting with lipid membranes and may have consequences for the adaptor function at the IM.

## Results

### Atg18 forms higher-order helical assemblies

In order to structurally characterize Atg18 oligomers and their interaction with lipid membranes, we recombinantly overexpressed *Saccharomyces cerevisiae* Atg18 and subsequently purified it under high-

salt conditions. Interestingly, once dialyzed into the low-salt buffer (100 mM KCl), Atg18 formed large supramolecular structures when imaged in negatively stained EM samples. Using cryo-EM of vitrified Atg18, we identified helical tubes of 260 Å diameter including a diamond-shape repeat pattern along the helix (Fig. 1A). We acquired cryo-EM micrographs of wildtype Atg18 (Atg18-WT) and subjected them to segmented helical image processing. 2D class averages of the helical tubes revealed individual Atg18 propeller disc-shaped structures including individual blade separation (Fig. 1B). The presence of helical layer lines beyond 1/10 Å$^{-1}$ in the corresponding Fourier transforms indicated an ordered helical repeat of the assembly (Fig. 1C). In an effort to improve the helical order, we imaged the previously characterized Atg18-P72A/R73A (PR72AA) mutant that has lost its ability to interact with Atg2[41]. The corresponding Fourier transforms supported the same structural organization of both helical assemblies, with an improved helical ordering in the Atg18-PR72AA mutant over the Atg18-WT, indicated by the higher diffracting layer line at 1/6 Å. The assemblies exhibited a pitch of ~100 Å with 5.1 helical units per turn. Using symmetry parameters of 19.4 Å helical rise and 70.0° rotation, we determined the helical structures to 3.8 Å and 3.3 Å, respectively (Fig. 1D–F; Suppl. Fig. 1A–E). Interestingly, Atg18-PR72AA tubes exhibited diameters of 210 Å whereas Atg18-WT tubes are slightly wider at 220 Å (Suppl. Fig. 1A–E). In both structures, the basic helical asymmetric unit contained four copies of Atg18 assembled in a diamond outline with an internal channel of 110 Å diameter.

When we inspected the cryo-EM maps, they showed the presence of expected density features at the determined resolution such as β-strand separation and side-chain densities (Fig. 1G). The observed features allowed for atomic model building of Atg18 into the cryo-EM density (Table 1, Fig. 1H). In comparison with the Atg18 crystal structure (PDB-ID 6KYB)[36], we found high structural overlap with only minor differences on the rotamer level (RMSD = 1.18 Å with 2471 atoms) and slightly extended density at the loop 7AB (Suppl. Fig. 2A–D). Residues beyond the β-propeller structural scaffold including the loops 4CD (157–221), 6CD (322–408) and parts of 7AB (446–457) were not resolved in the tube structures of Atg18-WT and Atg18-PR72AA, which is in agreement with both the AlphaFold2 prediction[48] (Uniprot ID P43601) and PDB-ID 6KYB. Although the Atg18-PR72AA tube structure was determined at a slightly better resolution when compared with the Atg18-WT (3.3 Å over 3.8 Å), the differences between the atomic models are only detectable as minor positional changes within the assembled tubular structure.

Common to both structures Atg18-PR72 AA and Atg18-WT are two major Atg18 interfaces in the assembly: first, inside the diamond, the interface is formed by two perpendicularly arranged Atg18 discs resulting in a "–|" or "T" configuration, and, second, the Atg18 interface between diamonds is formed by two Atg18 discs arranged in line giving rise to "– –" or "I" configuration (Fig. 2A–C, Suppl. Fig. 2E). The T-interface includes R285 and R286 of the FRRG motif that are in direct contact with blade 6 of the adjacent molecule (Fig. 2D). The I-interface is located between blade 3 and blade 4 due to a 180° disc rotation a pseudo dihedral symmetry including in a small tilt of the two discs with respect to each other. Notably, four T-interfaces are arranged in the characteristic lozenge pattern that was previously observed in the COPII coat formed by Sec31 tetramers (Fig. 2E).

### Tubular assembly is stabilized by FRRG motif

In order to better understand the lipid adaptor function of Atg18 in the context of the newly observed helical assembly, we mapped two binding sites of PI3P and PI(3,5)P$_2$ separated by the FRRG motif onto the presently determined Atg18 assembly structure (Fig. 3A, B). Interestingly, the lipid binding sites appear buried in the T-interface of the helical tube suggesting a stabilizing role of the involved residues. In reference to previous Atg18 mutations[34,37,38] that abolished binding to PI3P and PI(3,5)P$_2$, we replaced the positively charged arginines in the

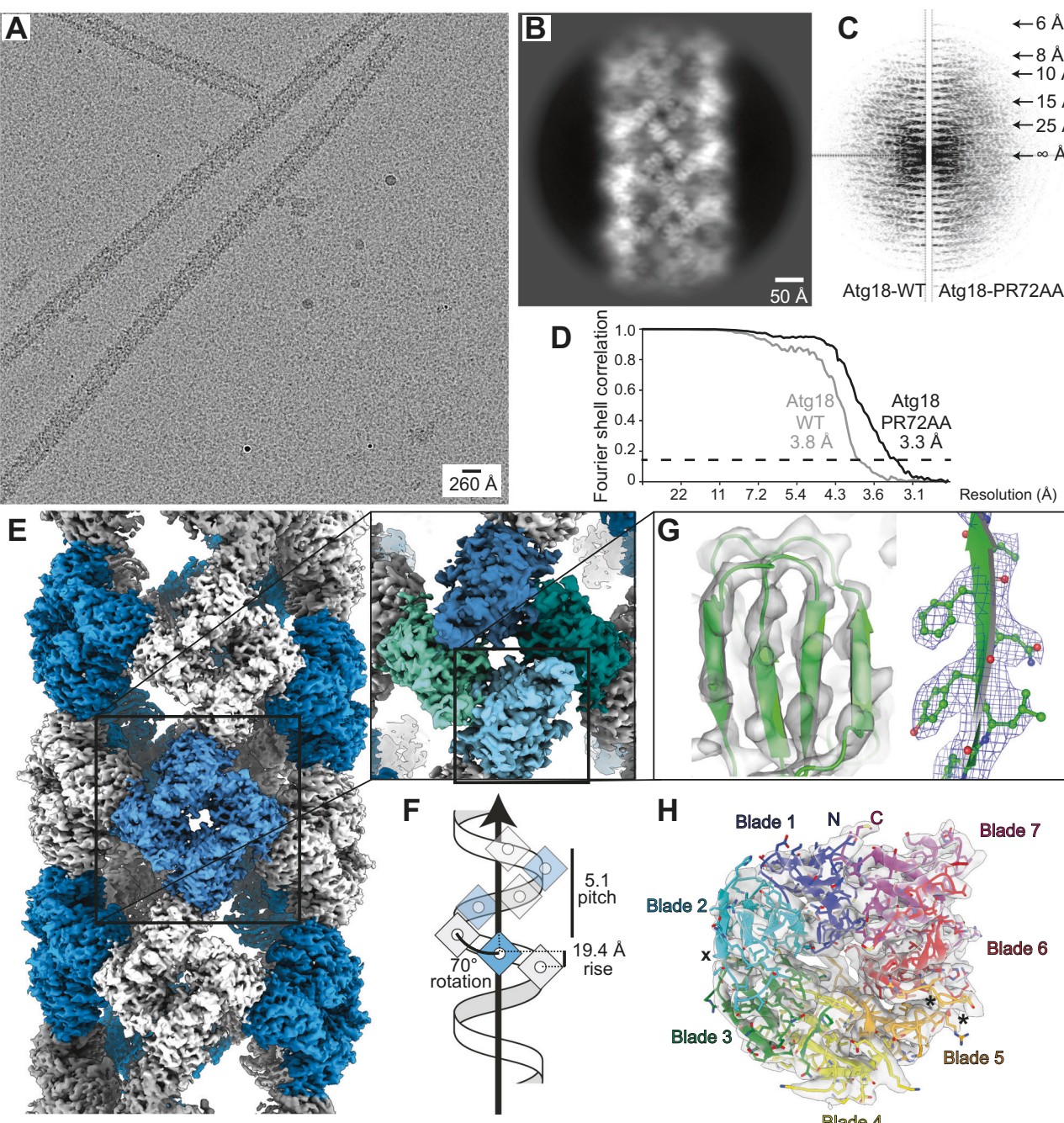

**Fig. 1 | Architecture of helical Atg18 assemblies and cryo-EM structure of Atg18-PR72AA. A** Micrograph of cryo-vitrified Atg18-P72A/R73A (PR72AA) tubes. Note, the grainy background is due to high concentrations of monomeric and oligomeric Atg18. The representative micrograph was taken from a dataset of 2380 movies. **B** Selected 2D classes and **C** the corresponding Fourier transform including helical layer lines. The panel is composed of the left half of Atg18-WT and right half of Atg18-PR72AA, the latter showing higher resolution diffraction in the power spectrum. **D** Fourier shell correlation threshold at 0.143 indicates resolutions of 3.8 and 3.3 Å for the 3D structures of Atg18-WT and Atg18-PR72AA, respectively. **E** 3D density of the helical tube formed by Atg18-PR72AA. The helical symmetry unit is illustrated with alternating blue/white colors (pitch: 100 Å, number of units per turn 5.1; helical rise: 19.4 Å, rotation 70.0°). One diamond-shaped asymmetric unit consists of four Atg18 molecules segmented in blue-green shades in the inset. **F** Schematic model of helical symmetry architecture. **G** Atomic model refined into the 3.3 Å map (shown as transparent surface) including density β-strand detail of blade 5 at the PIP-binding motif (I281–F284). **H** Top view of Atg18 ribbon model with the PR72AA mutation located by a cross (x) and the phosphoinositide binding sites in blade 5 indicated by asterisks (*).

FRRG motif with FGGG by neutral glycines. After purification, we found a complete solubilization of Atg18-FGGG over Atg18-WT tubes when subjected to a pelletation assay (Fig. 3C and Suppl. Fig. 3). Furthermore, Atg18-FGGG did not form tubes in comparison with Atg18-WT when imaged by negative stain EM. To further experimentally test the influence of the PIP-binding site on the tubular assembly, we added the soluble diC8-PIP derivates of PI3P, PI(3,5)P$_2$, and PI(4,5)P$_2$ in four-fold molar excess to Atg18-WT prior to low-salt buffer change and visualized these samples by negative stain EM. In contrast to the control-containing formed Atg18 tubes, only smaller particulate aggregates could be observed in the presence of PI3P, PI(3,5)P$_2$, and PI(4,5)P$_2$, respectively (Fig. 3D). Pelletation assays and subsequent separation by SDS-PAGE confirmed that the intensity of Atg18 bands moved to the supernatant fraction for PI(3,5)P$_2$ and PI(4,5)P$_2$ incubations (Fig. 3E). In

**Table 1 | Model report of four Atg18-PR72AA monomers (symmetry unit) built into the 3.3 Å density**

| Model | |
|---|---|
| Chains | 4 |
| Atoms | 10,139 (Hydrogens: 0) |
| Residues | Protein: 1304 nucleotide: 0 |
| Water | 0 |
| Ligands | 0 |
| Bonds (RMSD) | |
| Length (Å) (# > 4σ) | 0.004 (0) |
| Angles (°) (# > 4σ) | 0.995 (0) |
| MolProbity score | 1.74 |
| Clash score | 8.63 |
| Ramachandran Plot (%) | |
| Outliers | 0.00 |
| Allowed | 4.03 |
| Favored | 95.97 |
| Rama-Z (Ramachandran plot Z score, RMSD) | |
| whole (N = 1264) | 0.04 (0.24) |
| helix (N = 0) | --- (---) |
| sheet (N = 558) | 0.73 (0.23) |
| loop (N = 706) | −0.53 (0.23) |
| Rotamer outliers (%) | 0.00 |
| Cβ outliers (%) | 0.00 |
| Peptide plane (%) | |
| Cis proline/general | 0.0/0.0 |
| Twisted proline/general | 0.0/0.0 |
| CaBLAM outliers (%) | 1.72 |
| ADP (B-factors) | |
| Iso/Aniso (#) | 305/9834 |
| min/max/mean protein | 73.4/122.27/80.93 |

the case of PI3P, however, we observed a decrease in the pellet fraction in comparison with the untreated control as well as elongated aggregates in negative stain EM. Subsequently, we investigated whether preformed tubes were affected by the supplemented lipids. Interestingly, only $PI(3,5)P_2$ showed disassembled elongated aggregated tubes when visualized by negative stain EM (Fig. 3F), whereas the addition of PI3P or $PI(4,5)P_2$ did not affect the helical assemblies visibly. These observations are in line with the highest measured affinities of $PI(3,5)P_2$ to the FRRG site[39]. Together, these experiments show that the FRRG motif of Atg18 takes up a stabilizing role in the formation of the helical Atg18 assembly, which is further supported by binding studies of PI3P derivates that can prevent the assembly of Atg18 tubes and keep Atg18 soluble. The observed binding properties suggest that the helical Atg18 tubes will not form in the presence of PIP-containing membranes.

## Soluble Atg18 is composed of smaller oligomers

In order to further investigate the structures of soluble Atg18 fractions, we set out to determine the structural intermediates of the Atg18 assembly. As indicated by the pelletation assay of Atg18-WT, an equal-share fraction was not pelleted and remained in the supernatant. Using these preparations in high-salt buffer, we determined the single-particle cryo-EM structure of soluble Atg18-WT fractions. Due to the very small particle size, i.e., the Atg18 monomer corresponds to 55 kDa, we imaged over 1000 particles on a single cryo-micrograph (Fig. 4A). Classification analysis revealed separate views of single Atg18 displaying characteristic structural features of blade separation of the β-propeller (Fig. 4B). Alongside Atg18 monomer particles, we were also

able to classify a selection of Atg18 dimers (Fig. 4C). Comparison with projections of the T and I-interface dimers extracted from atomic model of the Atg18 tube matched some of the 2D classes (Fig. 4D). Moreover, we failed to reconstruct coherent dimer 3D structures of these Atg18 dimers presumably due to higher flexibility in solution than observed in the helical assembly. Nevertheless, using monomeric class members we determined the 3D structure of Atg18-WT at 4.8 Å resolution according to the FSC 0.143 criterion (Fig. 4E, F). Next, we successfully docked a single refined atomic model taken from the helically assembled Atg18 into the density of monomeric Atg18-WT (Fig. 4G). In accordance with the helical assembly structure, three major loop regions were not resolved and disordered in monomeric Atg18. Together, the structure determination of soluble Atg18 supports the occurrence of monomers and dimers in agreement with the T and I-interfaces observed in the helically assembled lattice structures.

## Visualization of membrane-associated Atg18 oligomers

To further investigate the binding mode of Atg18 to lipid membranes, we mixed soluble Atg18 with $PI_{3,5}P_2$-doped large unilamellar vesicles (LUVs). Next, we imaged the corresponding plunge-frozen samples by electron cryo-tomography. After acquiring a tilt series with the Volta phase plate, we reconstructed a total of 16 tomograms displaying strong low-resolution contrast. In comparison with common LUV controls that exhibited spherical vesicles of an average 30 nm diameter (Fig. 5A), the images showed tightly tethered deformed vesicles of square, pentagonal, or higher polygonal-like shapes with several 100 s nm long stretches of straight and parallel membrane paths next to other vesicles (Fig. 5B; Suppl. Movie 1). The membrane surfaces are typically coated by disc-shaped particles corresponding to Atg18 β-propellers in dimension. When membranes are aligned in parallel, we found connecting density that we interpreted as Atg18 bridging two juxtaposed bilayers.

In addition to the FRRG motif, Atg18 features an amphipathic helix in its 6CD loop that was shown to insert in the membrane and ultimately involved in scission[40]. In order to differentiate the two contributions of the binding properties of Atg18 to lipid membranes, we investigated samples of Atg18-WT mixed with LUVs lacking $PIP(3,5)P_2$ and an Atg18 6CD loop mutant (Atg18 s-loop) incapable of membrane insertion[40] mixed with LUVs containing $PIP(3,5)P_2$ (Suppl. Fig. 4A). The prepared Atg18 s-loop protein was folded correctly and formed dimers similar to Atg18-WT based on single-particle cryo-EM (Suppl. Fig. 4B–E). In cryo-tomograms obtained with 100% DOPC LUVs lacking $PIP(3,5)P_2$, we did not find an Atg18 membrane coat (Fig. 5C). Atg18 s-loop mixed with $PI(3,5)P_2$-doped LUVs formed coats on vesicles and bridged adjacent lipid membranes similarly to Atg18-WT as observed using cryo-electron tomography (Fig. 5D). The inter-membrane distance in the Atg18-WT and Atg18 s-loop samples was constant over parallel membrane stretches and very similar at ~80 Å, respectively, whereas no consistent inter-membrane spacing could be detected between undecorated 100% DOPC liposomes without Atg18 (Fig. 5E). Together, the binding experiments show that membrane association and inter-membrane alignment by Atg18 is primarily mediated through the FRRG-PIP interaction rather than the amphipathic helix of the 6CD loop.

In order to further analyze these membrane-associated structures, we applied membrane segmentation followed by membrane-guided particle picking using the PySeg package[49] to the cryo-electron tomograms (Fig. 6A, B). When generating rotational 2D class averages along the membrane plane from ~150,000 extracted subtomograms, some classes showed the presence of additional density between two 60 Å thick bilayers confirming the above-determined intra-membrane space distance of 80 Å (Fig. 6C). Other classes containing a single membrane only or a second more blurred membrane were excluded from further image processing. Further 3D subclassification resulted in a structure of two S-shaped densities packed against each other made up of eight disc-shaped densities with ~50 Å diameter each (Fig. 6D, E).

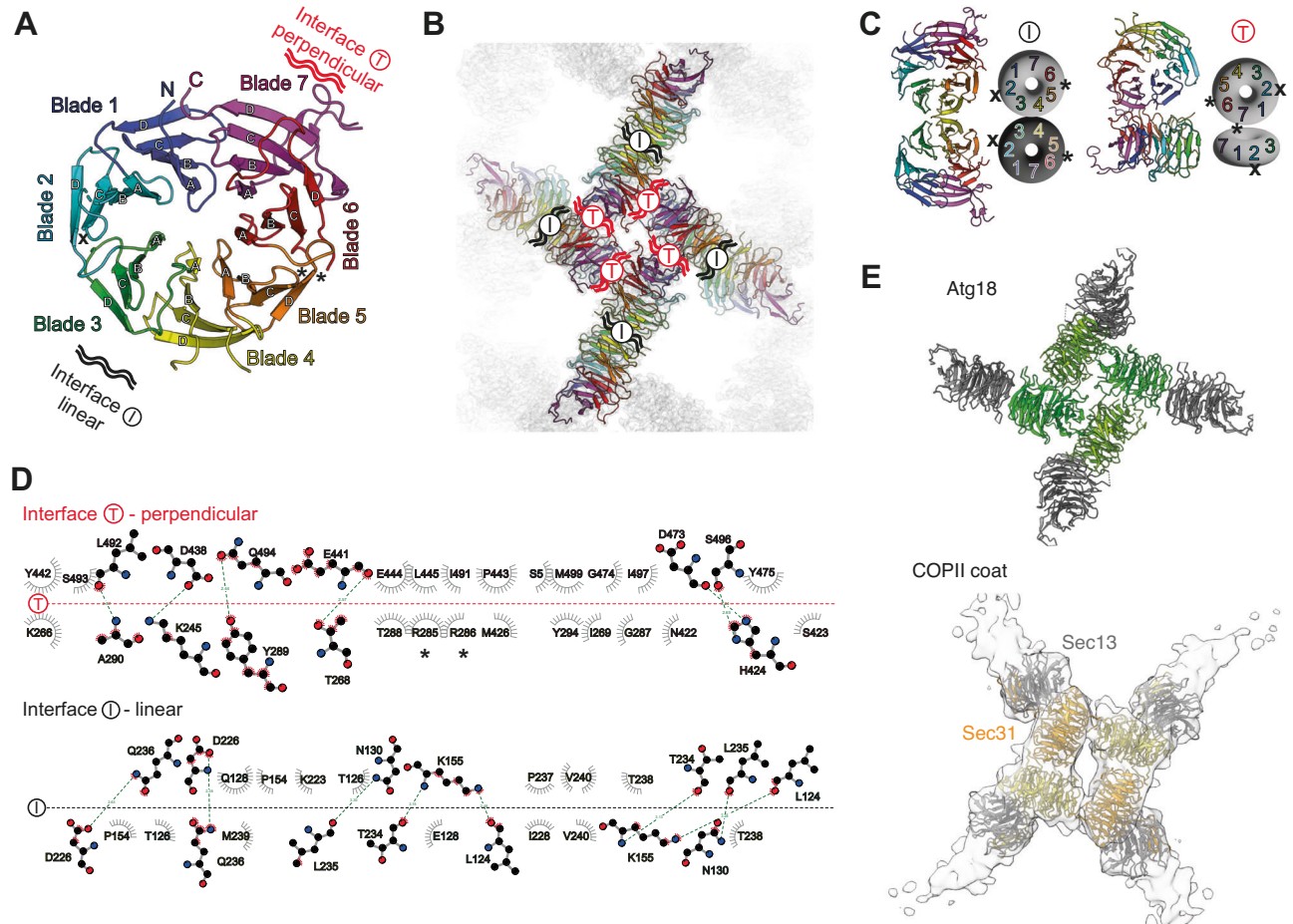

**Fig. 2 | Atg18 assembly interfaces present in the helical tubes. A** T-Interface (upper right/red) and I-interface (lower left/black) are indicated on the same cartoon color code as in Fig. 1. Location of P72/R73 mutation in loop 2BC is indicated by (x). The FRRG motif is indicated by asterisks (*). **B** Refined atomic models placed inside 3.3 Å cryo-EM density (transparent background) of Atg18-PR72AA helical assembly. Color code is identical to **A**. T and I-interfaces are indicated in black and red, respectively. **C** Schematic ribbon presentation of I and T-interface dimers, P72/R73 (x), and the FRRG motif (*) are indicated. **D** Projected side-chain dimer interaction plot (DIMPLOT) of the T (top) and I (bottom) interfaces. H-bridges and their atom distances indicated as dashes, van der Waals interactions are indicated as shielded residue symbols. FRRG motif indicated as asterisks (*). **E** The fourfold arrangement of T-interfaces is also found in the Sec31/Sec13 COPII coat.

A total of 8300 subtomograms resulted in a structure at 26 Å estimated by mask-less FDR-FSC[50] (Suppl. Fig. 5). In the density, we found Atg18 molecules in four pairs of dimers with a slightly twisted interface matching the observed I-interface of the tubular Atg18 assembly determined above (Fig. 6F). Notably in the cryo-EM density, the basic Atg18 dimer unit was found to assume a 45° tilt angle with respect to both membrane planes, which could be independently confirmed by measurements in raw tomogram slices (Fig. 6G). The rigid body fit of the Atg18 dimer was consistent with the location of the PI3P binding sites as well as the 6CD loop facing opposing membrane bilayers (Fig. 6H). Moreover, we analyzed those subtomograms that did not contain two juxtaposed bilayers and processed Atg18 monolayer coats. Two resulting classes could be further averaged at a resolution of 28 Å and found that 70% of Atg18 β-propeller particles adopted a similar 45° angle with respect to the bilayer (Suppl. Fig. 6). The determined subtomogram structure reveals that tilted Atg18 dimers in I-configuration can establish contact between two opposing bilayers and can further align longer stretches of two opposing lipid membranes in constant distance of ~80 Å.

### Models of Atg2-Atg18 complex on lipid membranes
In order to further interpret the experimentally obtained subtomogram average structures in the context of the Atg2 complex, we computed the protein complex prediction of N-terminally truncated Atg2

(541–1592) with Atg18 using the AlphaFold-multimer approach[51] (Suppl. Fig. 7A). The lowest energy structure prediction of the Atg2 (541–1592)-Atg18 complex showed two major interfaces that are consistent with previously determined binding interfaces while leaving the I-dimer interface available for other Atg18 molecules (Suppl. Fig. 7B): first, Atg18-PR72AA had been mutated to disrupt Atg2 binding[41] and second, a WIR motif-containing peptide of human ATG2A had been crystallized with WIPI3[52]. This x-Φ-x-Φ-x-x-x-φ-F PROPPIN interaction motif identified in human ATG2A corresponds to residues 921–938 in Atg2 that bind between Atg18's blade 1 and 2 (Suppl. Fig. 7C). The full-length Atg2 model is available from the Alphafold-EBI databank (Uniprot ID P53855). The Atg2 molecule consists of a 200 Å long β-helix cylinder with a hydrophobic channel inside predicted at high confidence with accessory loops and α-helices predicted at lower confidence. When extending the truncated Atg2 (541–1592) model with the predicted full-length Atg2 and the determined Atg18 dimer, one can predict the membrane-bound structure of the 2xAtg18-Atg2 complex bridging opposite membranes[27] (Suppl. Fig. 7D, E). In both rigidly extended complex models with tilted Atg18 dimers forming the structural scaffold, Atg2 assumes a tilted orientation with respect to the lipid membrane.

## Discussion
In order to study the structural role of the Atg18 autophagy membrane adapter, we determined the structures of tubular and soluble Atg18

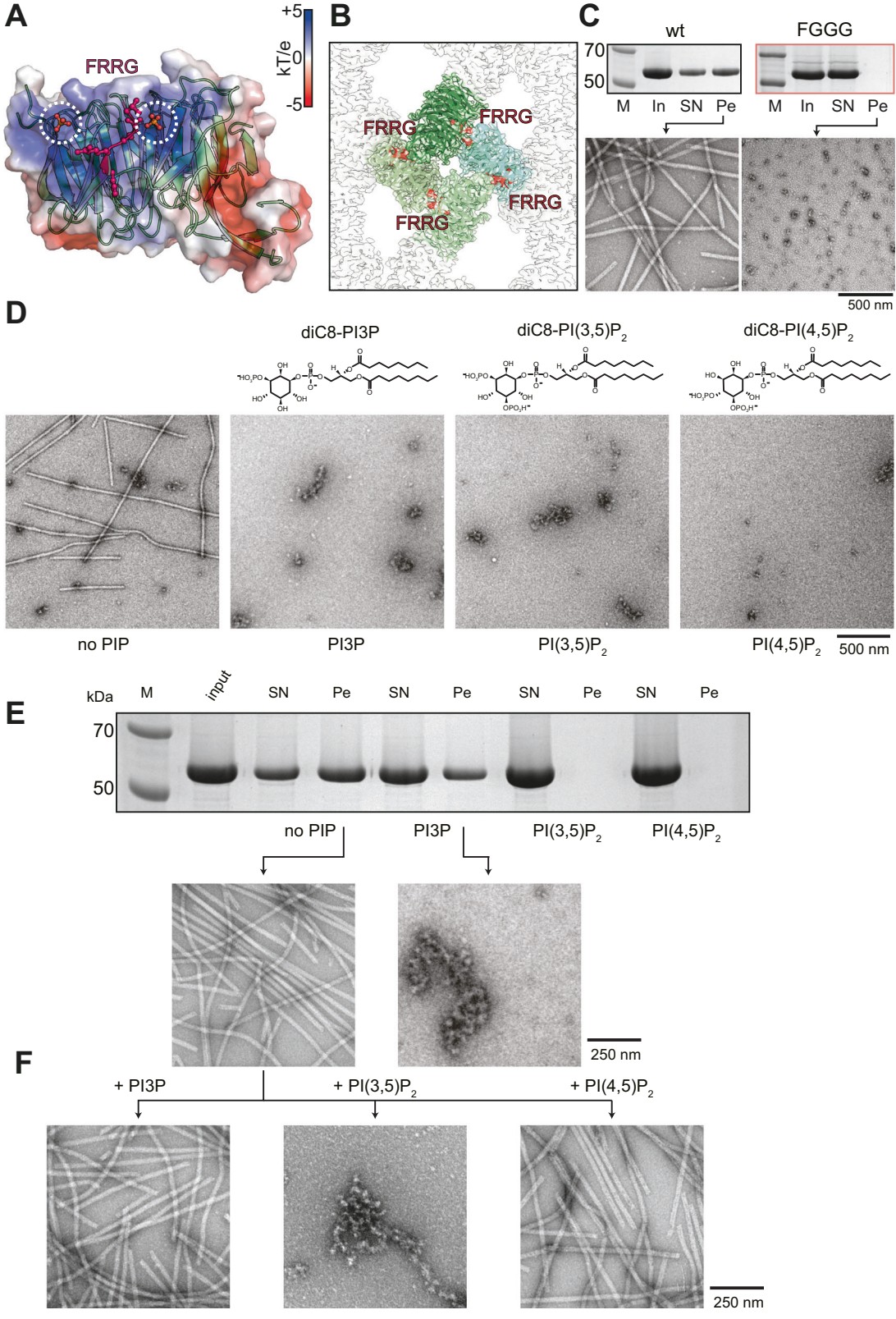

oligomers using high-resolution cryo-EM. Common to these Atg18 structures are two principal oligomeric binding modes that are mediated by so-called T and I-interfaces, respectively. The FRRG motif that is known to bind PIP constitutes a critical part of the T-interface thereby enabling the formation of a large tubular lozenge Atg18 lattice. When Atg18 is reconstituted with lipid membranes containing (PI) 3,5P$_2$, subtomogram averaging reveals Atg18 oligomers including an elongated tilted Atg18 dimer at the core, which is capable of juxtaposing two opposite membrane bilayers. The autophagic membrane adapter Atg18 reveals an unexpected structural plasticity in multiple modes of oligomeric organization.

Initial structural characterization of purified Atg18 revealed the formation of higher-order helical assemblies under low-salt conditions (Fig. 1). The determined cryo-EM structures of Atg18-WT and Atg18-

**Fig. 3 | PIP binding by the FRRG motif prevents Atg18 tube formation. A** Atg18-PR72AA refined atomic model with surface colored by electrostatics, FRRG motif in pink, and phosphate binding sites from aligned PDB-ID 5LTD. **B** Diamond-shaped tetramer unit (green) with FRRG motifs (red) and their adjacent phosphate binding sites, which are buried in Atg18 tubes. **C** Pelletation assay (top) and negative stain of pellet fractions from Atg18-WT (left) and Atg18-FGGG (right) (M = marker, In = input, SN = supernatant, Pe = pellet). **D** Top: Chemical formulas of soluble PIP derivates diC8-PI3P, diC8-PI(3,5)P₂ and diC8-PI(4,5)P₂ used in binding experiments.

Bottom: negative stain electron micrographs of Atg18 nontreated or treated with 4x molar excess of the indicated PIP derivate preventing tube formation. **E** Pelletation assay with corresponding SDS-PAGE of the supernatant (SN) and the pellet (Pe) fraction and the negative stain EM micrographs of two pellets. **F** Tubes from the negative control were treated with PIP derivates. Only PI(3,5)P₂ resulted in tube disassembly. Source data are provided as a Source Data file. Displayed micrographs are representative examples from screening sessions with dozens of negative stain micrographs.

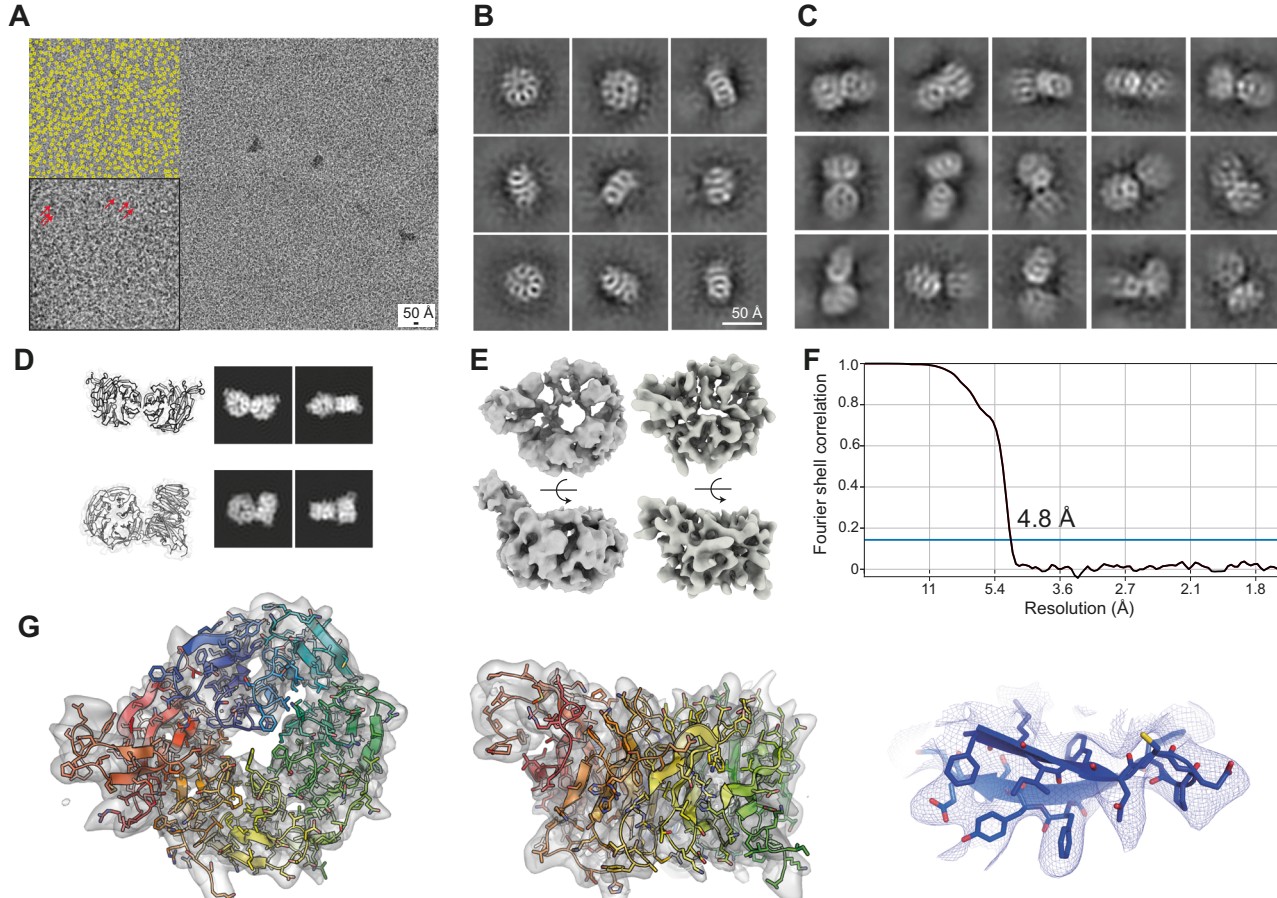

**Fig. 4 | Single-particle cryo-EM structures of soluble Atg18. A** Typical cryo-micrograph of Atg18-WT with over 1000 particles present (yellow circles in inset). Selected individual molecules are also indicated by red arrows. This representative micrograph was selected from a dataset of 5117 movies. **B** Representative 2D classes of single Atg18 particles with features of individual blades of the β-propeller. **C** Representative 2D classes of Atg18 dimers from the same dataset. **D** 3D views and 2D projections of I-interface dimer (top) and T-interface dimer (bottom) for

comparison with 2D classes in panel C. Some experimental classes are consistent with the packing of I and T-interface dimers observed in the tubular assembly. **E** Ab initio initial 3D model (left) and final 3D structure (right) of Atg18-WT. **F** Fourier shell correlation at the 0.143 threshold indicated global resolution of 4.8 Å. **G** Fitted atomic model taken from Atg18-PR72AA tube structure superimposed with the determined 3D structure of monomer Atg18.

PR72AA constitute two main propeller interactions mediated by the T and I-interface between neighboring monomers (Fig. 2). Analysis of crystal symmetry contacts in a recent X-ray structure revealed identical T and I-interfaces in low-salt buffer, even though the protein was trypsinized to remove the disordered loop regions[36]. Proteolytically truncated Atg18 forms a tightly packed lattice suitable for 3D crystallization but does not form a hollow cylindrical lozenge lattice built from diamond-shaped tetramers, which we observed for full-length Atg18 in the electron micrographs. Interestingly, the outer membrane-trafficking COPII coat of Sec13-31 displays a very similar structurally related diamond arrangement built from four T-shaped β-propeller units forming a scaffold around lipid membrane tubules[53] (EMD-11194, PDB-ID 6GZ6) (Fig. 2, Suppl. Fig. 8). Similar roles of coat formation around lipids can also be envisioned for the observed Atg18 tubes in particular as previous studies showed that Atg18 is capable of

tubulating GUVs[40]. Moreover, recently two groups identified Atg18 independently as a component of the CROP complex in a membrane-associated retromer complex[54,55]. This architectural conservation of proteinaceous coats suggests functional parallels between different modules of COPII and autophagy trafficking.

The structural similarity of Atg18 tubes to COPII outer membrane coat is a particularly noteworthy observation considering that COPII vesicles have been shown to deliver lipids to the autophagosome IM[20]. Moreover, phosphorylation states of COPII components affect autophagosome abundance in the cell presumably by modulating coat assembly[19]. Similarly, when Atg18 was incubated in the presence PIP, we observed the disassembly of the tubular structures into soluble structures suggesting a functional precursor role of the resolved tubular assemblies. In fact, the T-interface buries the PI3P/PI(3,5)P₂ FRRG binding interface and shows direct competition between PIP binding

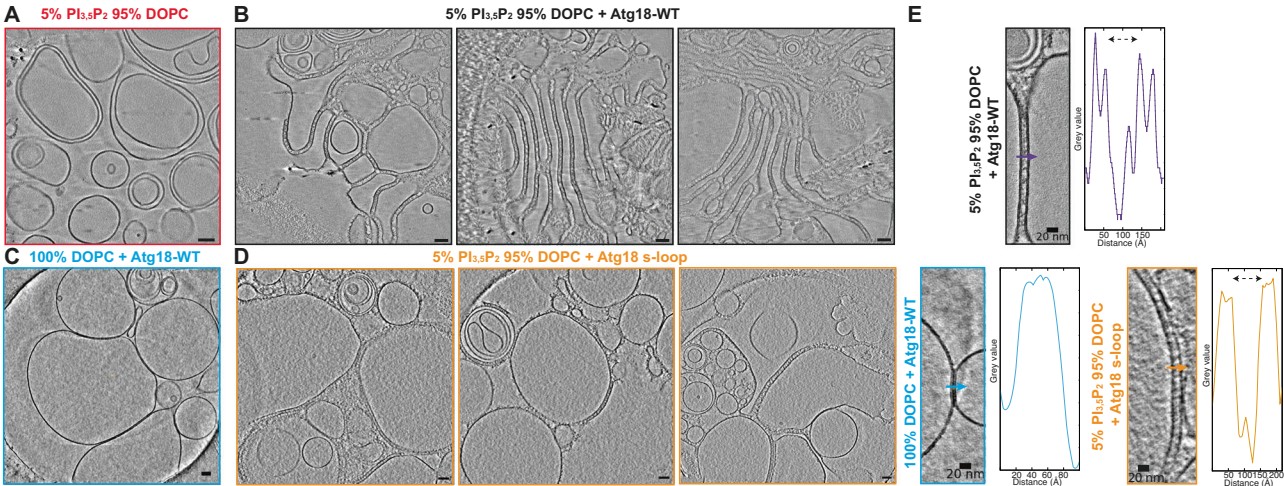

**Fig. 5 | Cryo-electron tomography of membrane-interacting Atg18. A** Large unilamellar vesicles (LUVs) (5% $PI_{3,5}P_2$, 95% DOPC) represent the negative control in the absence of any Atg18. **B** Atg18-WT affected the curvature of LUVs (5% $PI_{3,5}P_2$, 95% DOPC), formed coats, and tethered adjacent membranes. **C** LUVs made of 100% DOPC lacking $PI_{3,5}P_2$ in presence of Atg18-WT showed no Atg18 association. **D** LUVs (5% $PI_{3,5}P_2$, 95% DOPC) incubated with Atg18 scrambled 6CD loop (s-loop) showed similar binding features comparable with Atg18-WT experiments. Scale bar 50 nm (**A–D**). **E** Greyscale density profiles along indicated vectors of Atg18-WT (purple), the $PI_{3,5}P_2$ negative control (blue), and the s-loop variant (orange) showed similar inter-membrane distance of protein-bound Atg18-WT and Atg18 s-loop. In the $PI_{3,5}P_2$ control, however, in the absence of a tethering Atg18 protein, the two vesicles directly contact each other (blue). Inter-membrane distances are indicated by dashed arrows. Shown tomogram slices were representative of data collections from a dozen tomograms for each sample.

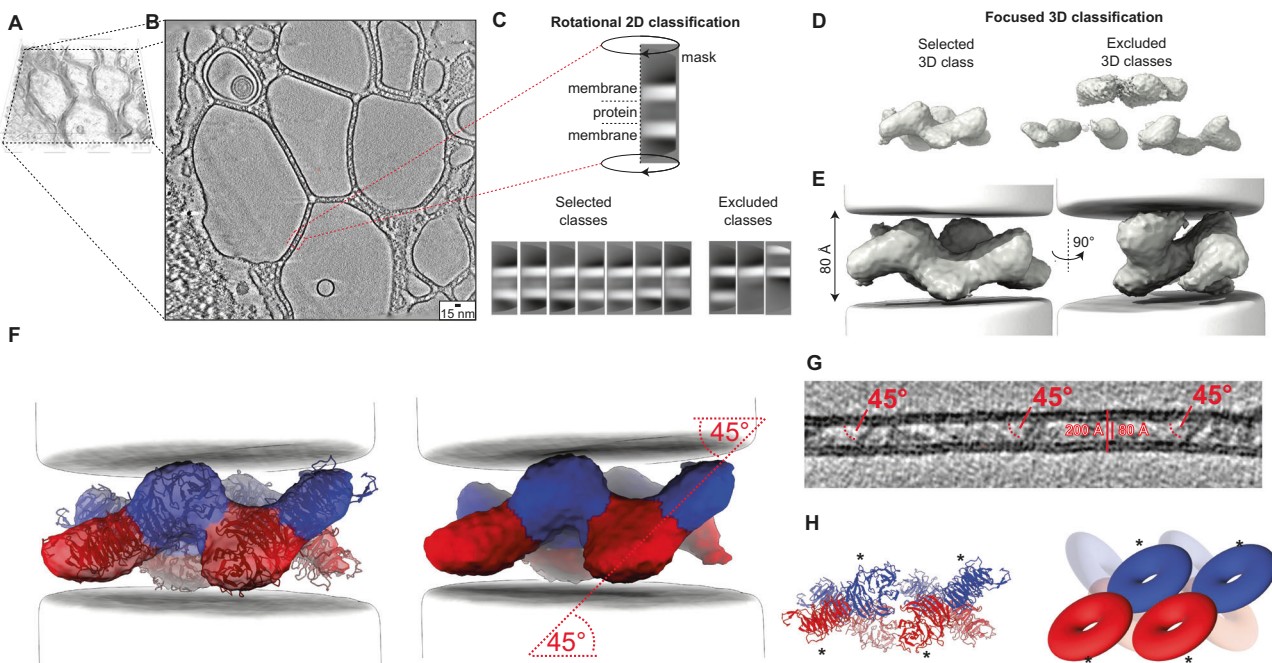

**Fig. 6 | Cryo-electron tomography analysis of membrane-bound Atg18-WT oligomers. A** Volume density rendering of a reconstructed tomogram containing Atg18-WT and PI(3,5)$P_2$-containing large unilamellar vesicles (LUVs). **B** Central slice of the corresponding tomogram acquired with Volta phase plate (averaged five slices to display increased contrast). **C** Examples of rotational 2D classification after subvolume extraction of double-membrane subtomograms. The 60 Å thick membrane bilayers are spaced apart by 80 Å. **D** Initial subvolume average dominated by membrane density contributions. **E** Focused 3D classes after membrane subtraction reveal protein densities. **F** Refined structure of the selected 3D class (C1 symmetry) shows organized stacking of eight disc-shaped molecules. Density fit with four Atg18-WT dimers formed by I-interface previously observed in Atg18 tubular assemblies and soluble dimers. The Atg18 molecule within the dimer is colored blue and red, respectively. **G** Slice through reconstructed tomogram reveals Atg18 dimer stacking with the ~45° angle relative to the membrane bilayer planes. **H** Cartoon representation of docked molecules with FRRG motif indicated by asterisks (*).

and tube formation (Fig. 3). Therefore, our data suggest that the characterized Atg18 tubes act functionally prior to engaging in its adaptor role at the phagophore. In yeast, filamentous assemblies have been shown to accumulate in light microscopic punctae under stress conditions thereby regulating the activity of metabolic enzymes[56].

Therefore, a similar storage role may also be envisioned for the observed Atg18 assemblies.

Many other components of the core autophagy machinery such as Atg9 and Atg13 were shown to be activated and de-activated by chemical modification of phosphorylation[57,58]. More specifically for Atg18,

it was shown for *Pichia pastoris* that Atg18 itself is subject to phospho-regulation by two distinct sites in loops 6CD and 7AB, both reducing PIP-binding affinities[59]. Under normal conditions in the cell, Atg18 is phosphorylated in loop 6CD and thereby the negative charge prevents the insertion of the amphipathic helix into the membrane[40]. Similarly, Atg18 phosphorylations in loop 7AB can be expected to affect the stability of the tubular assembly, in particular, as the higher resolution structure of Atg2-binding deficient mutant Atg18-PR72AA tubes suggested an increased rigidity over Atg18-WT. Conserved regulatory phosphorylation mechanisms of Atg18 will affect the tube stability and may prime smaller Atg18 oligomers to bind PIP-containing membranes.

Yeast and mammalian autophagosomes exhibit distinct PIP asymmetries, i.e. PI3P is present at the IM while almost absent in the ER membrane[60] whereas $PI_{3,5}P_2$ is found localized in late endosomes, autophagosomes, and vacuoles[38]. The differential binding affinities of different PIP derivatives to Atg18 have been biochemically characterized and quantified before in detail[39]. In line with the highest affinity, we found that $PI_{3,5}P_2$ binding leads to disassembly of the helical tubes. The lower binding affinity of PI3P may require additional membrane tethers like the related PROPPIN Atg21[43]. Binding to Atg2 through P72/R73 in loop 7AB[41] may contribute further to the destabilization of the tubes in order to assist in the targeted localization of Atg18 to phagophore membranes.

When we investigated soluble Atg18 fractions at high-salt conditions by single-particle cryo-EM, we confirmed the presence of monomers and dimers in line with previous native mass spectrometry characterizations[35] (Fig. 4). Mass spectrometry cross-linking identified residues in loops 4 BC and 6CD engaged in Atg18 oligomer interactions[35] consistent with the here determined T and I-interface. Furthermore, the study also revealed that the interaction patterns between Atg18 oligomers change in the presence and absence of liposomes[35]. In support, when we incubated Atg18 with PIP-containing membranes, we observed the decoration of Atg18 β-propellers on lipid bilayers and the bridging of two bilayers leading to a parallel alignment of two juxtaposed bilayers (Fig. 5). As previously demonstrated by fluorescence microscopy, biophysical, and biochemical characterizations as well as molecular dynamics simulations[35,39,40], Atg18 interacts with lipid membranes through the FRRG motif and the amphipathic 6CD loop in a PIP-dependent or independent manner, respectively. We found membrane binding of Atg18 in cryo-electron tomograms to be $PI_{3,5}P_2$-dependent whereas the amphipathic 6CD loop alone was insufficient to associate Atg18-WT to PIP-devoid membranes. Conversely, we used the well-characterized mutant Atg18 s-loop[40] to prevent 6CD loop-dependent membrane insertion. In conclusion, Atg18 membrane association is largely PIP-dependent and the 6CD loop alone is insufficient to anchor the protein to the membrane. Due to the demonstrated importance of the 6CD amphipathic loop[40], we putatively assign other membrane interaction roles, e.g., the ability to complete membrane fission, to this part of the protein. The determination of subtomogram average structures between juxtaposed bilayers revealed decorated dimers and tetramers oriented at an angle of 45° with respect to the membrane plane. The fitted Atg18 dimers lack the interaction through the T-interface due to PIP-binding (Fig. 6). Together, Atg18 displays an unexpected structural plasticity of different multimer assemblies depending on buffer condition as well as binding partners, which is mechanistically accomplished by two distinct binding interfaces.

In order to put the determined Atg18 dimer structure bound to the lipid membrane in the context of the Atg2 complex, we computed Alphafold models of the Atg18-Atg2 complex (Suppl. Fig. 7). Here, Atg18 formed two contacts with Atg2 through P72/R73 (loop 2BC) and blade 1/2[41,52]. In an expanded Atg2-Atg18 dimer complex, the I-interface as well as the FRRG motif in blade 5 including loop 6CD are spatially accessible to engage in oligomer and membrane binding. In the Atg18 scaffolding configuration, the rod-shaped 20-nm long Atg2 molecule assumes an angle of approx. 45° with respect to the membrane bilayer bridging membrane distance of ~10 nm. In vitro reconstitutions of Atg2 and ATG2A increased lipid transfer across vesicles in the presence of Atg18 and WIPI1/4, respectively[27,28]. Therefore, it is tempting to speculate that the observed Atg18 dimers participate in Atg2 positioning, either by increasing the number of membrane contacts at the isolation membrane or directly bridging isolation membrane and ER membrane.

WD40 domains constitute a universal protein interaction scaffold within multiprotein complexes involved in many different biological structures such as histone and ubiquitin-binding modules as well as membrane-trafficking complexes[61–64]. In autophagy trafficking Atg18, Atg21, and Hsv2 as well as WIPI1-4 belong to the family of PROPPINs of yeast and mammalia, respectively. Given the ability of Atg18 to form oligomeric structures, it is conceivable that this structural property is conserved in higher eukaryotes. Upon starvation and co-transfection of GFP-WIPI1 in human cells, large cellular punctate structures including lasso filaments were observed by fluorescence microscopy[46] compatible with the here observed Atg18 tubes. In higher eukaryotes, ATG2A is known to interact with WIPI1 and WIPI4 whereas WIPI2 contacts ATG16L[47,65] another component of the LC3 lipidation machinery. In yeast, Atg18 and Atg21 co-localize to the PAS and Atg18 forms a complex with Atg2 whereas Atg21 contacts Atg16 of the Atg8 lipidation machinery[43]. Given the complementary functions and overlapping cellular locations of the WIPI proteins, it will require further investigation whether WIPI proteins form hetero-oligomeric complexes in addition to the observed homo-oligomeric Atg18 structures. Given the unexpected and characterized structural plasticity of yeast Atg18, future research on the entire WIPI family proteins is warranted to characterize their structural states and functional relevance and thereby delineate Atg18's contributions to autophagy and membrane-trafficking processes.

## Methods

### Protein expression and purification

N-6xHis-tagged Atg18 (Uniprot ID P43601), Atg18-FGGG, Atg18-PR72AA, and Atg18 s-loop included in pEXP5-NT/TOPO vectors with ampicillin resistance were recombinantly expressed in *Escherichia coli* BL21(DE3) cells. Briefly, after cell growth to $OD_{600nm}$ of 0.6 the temperature was lowered to 20 °C, and expression was induced by 0.2 mM IPTG overnight. Cells were harvested at $6000 \times g$ and resuspended in lysis buffer (50 mM TRIS, 500 mM NaCl, 10 mM $KPO_4$, 10 mM imidazole, 1 mM DTT, 0.5 µg/ml DNaseI, 0.1% Triton X-100, pH 7.5). Cells were lysed by homogenization in a fluidizer (EmulsiFlex-C3, Avestin or CF1 cell disrupter, Constant Systems) using 3 passes. Cell debris was cleared by centrifugation at $40,000 \times g$ for 45 min at 4 °C and supernatant was applied to His-tag affinity purification using Ni-NTA agarose (beads or 5 ml column, Qiagen) in buffer A (50 mM TRIS, 500 mM NaCl, 10 mM $KPO_4$, 10 mM imidazole, 1 mM DTT, pH 7.5) and eluted in the same buffer supplemented with 250 mM imidazole. The N-6xHis-tag was cleaved with home-prepared TEV protease, imidazol was removed overnight by dialysis and the cleaved 6xHis-tag was removed by a second round of His-tag affinity purification. The flow-through and wash fractions were pooled and concentrated to ~1 mg/ml using a 30 kDa cutoff spin concentrator (Amicon) before loading to a pre-equilibrated (50 mM TRIS, 500 mM NaCl, 10 mM $KPO_4$, 1 mM DTT, pH 7.5) gel filtration column (HiLoad 16/600 Superdex 200 pg, GE Healthcare). Peak fractions containing Atg18 were pooled and concentrated using a 30 kDa cutoff spin concentrator to 8–10 mg/ml. Protein aliquots were snap-frozen in liquid nitrogen and stored at −80 °C.

### Atg18 tube formation

Atg18 aliquots were thawed and diluted to 100 µM with tube buffer (20 mM HEPES, 100 mM KAc, 1 mM DTT, pH 7.2) and dialyzed against

**Table 2 | Acquired cryo-EM datasets**

| Dataset | Atg18-WT | Atg18-PR72AA | Atg18-WT | Atg18 + lipids |
|---|---|---|---|---|
| Processing type | Helical | Helical | Single particle | Subtomogram averaging |
| Microscope | Titan Krios | Titan Krios | Talos Arctica | Titan Krios |
| Detector | Gatan K2 Summit | Gatan K2 Summit | Gatan K3 Bioquantum | Gatan K2 Summit |
| Physical pixel size (Å) | 1.35 | 1.35 | 0.8389 | 2.129 |
| Total fluence ($e^-/Å^2$) | 90 | 90 | 70 | 120 |
| #Micrographs (accepted %) | 1812 (93) | 2380 (98) | 5117 (98) | 704 (16 Tomograms) (100) |
| #Particles final | 133,981 | 87,223 | 124,369 | 8,300 |
| Resolution (Å) | FSC(0.143) = 3.8 | FSC(0.143) = 3.3 | FSC(0.143) = 4.8 | FDR-FSC(1%) = 26 |
| Symmetry applied | Helical (rise 19.4 Å rotation 70.0°) | Helical (rise 19.4 Å rotation 70.0°) | None | None |
| Atomic model | Rigid body fit from Atg18-PR72AA | Refined (Table 1) | Rigid body fit of monomer from Atg18-PR72AA | Rigid body fit of dimer from Atg18-PR72AA |

the same buffer overnight (10 kDa membrane, ThermoScientific or Qiagen) at room temperature. Precipitates were cleared by centrifugation at $2000 \times g$ for 10 min and the pellet was disposed. To investigate tube formation and pelletation behavior of Atg18 variants or to enrich Atg18 tubes, 100–150 µl of sample were centrifuged for 120 min at $50,000 \times g$ and 18 °C. The supernatant was carefully removed and the pellet rinsed once with 150 µl tube buffer without detaching the pellet from the tube bottom. For semi-quantitative analysis of the pelletation, the pellet was resuspended in the same volume as the input to the pelletation, and identical volumes of input, supernatant, and pellet were analyzed by SDS-PAGE. For EM or other experiments, the pellet was resuspended in the desired volume.

### Soluble lipid and liposome binding experiments
To test the effect of soluble lipids on Atg18 tube formation, diC8-PI3P and diC8-PI$_{3,5}$P$_2$ (Echelon) were reconstituted in water (1 mM final concentration) and added in 4× molar excess to Atg18 in gel filtration buffer prior to the dialysis into tube buffer. As a control, the same procedure was performed with water instead of reconstituted lipids. To test the effect of soluble lipids on preformed Atg18 tubes, the soluble lipids in water were added to preformed Atg18 tubes in 4× molar excess. As a control, the same procedure was performed with water instead of reconstituted lipids for the preformed Atg18 tubes. LUVs were prepared with a lipid content of 95% DOPC and 5% DO-PI$_{3,5}$P$_2$ (Avanti Polar Lipids). DOPC was dissolved in chloroform whereas DO-PI$_{3,5}$P$_2$ was chloroform:methanol:water (20:9:1 v/v/v). Lipid films were generated in a glass vial by evaporating the solvent under a gentle stream of argon followed by vacuum desiccation. The lipid films were rehydrated in LUV buffer (20 mM HEPES, 50 mM NaCl, pH 7.1) for 60 min at 37 °C, vortexed for 30–60 s, and transferred to an Eppendorf tube at a final concentration of 1 mg/ml. LUVs were formed by three freeze-thaw cycles in liquid nitrogen and a 30 °C water bath. LUVs were snap-frozen in liquid nitrogen and stored at −20 °C until further use.

### Negative stain electron microscopy
300 mesh Cu grids with predeposited continuous carbon were purchased from Electron Microscopy Science or a homemade 5–10 nm thick carbon film (Leica EM ACE600) was floated manually onto 400 mesh Cu/Rh grids (Plano GmbH). Grids were glow discharged in air plasma (0.45 mbar) for 45 s (Pelco EasiGlow) before use. A total of 3.5 µl of the respective sample was applied to the carbon side of the grid, adsorbed for 2 min, and gently blotted from the side of the grid with filter paper (Whatman 1). Grids were washed twice with drops of sample buffer followed by immediate blotting. Staining of the adsorbed sample was performed with 2% uranyl acetate solution twice, followed by blotting and drying for at least 15 min before imaging. Images were taken on a FEI Morgagni 268 operated at 100 kV equipped

with a 1k CCD camera (Soft Imaging Solutions), on a Philips Biotwin CM120 operated at 120 kV equipped with an identical camera, or on a FEI Tecnai Spirit operated at 120 kV equipped with a 4k CCD camera (Gatan UltraScan 4000).

### Electron cryo-microscopy
A total of 3 µl 20 µM Atg18 tube solution in tube buffer (20 mM HEPES, 100 mM KAc, 1 mM DTT, pH 7.2) was applied to freshly glow discharged UltrAuFoil R1.2/1.3 300 mesh grids (Quantifoil) and plunge-frozen in liquid ethane using a Leica EM GP (18 °C, 99% humidity, 1.5 s single blot) or a ThermoFisher Vitrobot Mark IV (18 °C, 100% humidity, 1 s single blot, blot force 6). A total of 3.6 µl 18 µM monomeric Atg18-WT or Atg18 s-loop solution in high-salt buffer (50 mM TRIS, 500 mM NaCl, 10 mM KPO$_4$, 1 mM DTT, pH 7.5) was applied to freshly glow discharged R1.2/1.3 Cu200 grids (Quantifoil), blotted for 7 s (blot force −5, 4 °C, 95% humidity) and immediately plunge-frozen in liquid ethane in a Vitrobot Mark IV (ThermoFisher). LUVs were prepared as described above. For tomography of Atg18 on LUVs, 1 µl Atg18 in 20 mM HEPES, 150 mM NaCl, pH 7.2 was added to 9 µl of LUVs in LUV buffer. The final Atg18 concentration was 3 µM, LUVs at 0.1 mg/ml. A total of 3 µl of the solution was applied to freshly glow discharged R2.2 Cu300 (Atg18-WT) or R1.2/1.3 Cu200 (Atg18 s-loop) grids (Quantifoil), blotted from the back side for 1.2 s (Atg18-WT) or 6.0 s (Atg18 s-loop) at 20 °C and 99% humidity and immediately plunge-frozen in liquid ethane using a Leica EM GP. A total of 16 tomograms were recorded in focus with Volta Phase Plate induced phase shifts in dose symmetric scheme starting at 0° in ±3° increments to 60° with 3e$^-$/Å$^2$/tilt fluence for Atg18-WT on a 300 kV Titan Krios (ThermoFisher) equipped with a K2 summit direct electron detector (Gatan) using SerialEM[66]. A total of 30 tomograms were recorded without phase plate with approx. −3 µm defocus for Atg18-Sloop with identical dose distribution on a 300 kV Titan Krios equipped with a Falcon IV detector (ThermoFisher). Single-particle data was recorded on a 200 kV Talos Arctica (ThermoFisher) equipped with a filtered K3 Bioquantum detector. Data collection parameters are summarized in Table 2.

### Cryo-EM image processing
Micrographs containing Atg18 tubes and soluble monomer/oligomers were processed in CryoSPARC v3.3.1[67–69]. Movies were patch motion corrected, dose weighted, and binned to physical pixel size. Patch CTF estimation was performed and micrographs with poor CTF fit resolution >6 Å were excluded (see % used in Table 2). Helical tubes were automatically traced (diameter 260 Å, 0.3 (78 Å) step distance), segmented in 400 px/540 Å boxes, and 2D classified. High-resolution 2D classes containing propeller blades were selected. Initially, a symmetry search was performed in the SPRING software package using SEGCLASSRECONSTRUCT[70], resulting in symmetry values for pitch

and number of units per turn of 99.8 Å and 5.1. The selected classes were also subjected to helical refinement starting with a featureless cylinder without symmetry parameters, resulting in an ~5 Å 3D reconstruction. Symmetry search in CryoSPARC yielded a global minimum at helical rotation of 70° and rise of 19.4 Å. Helical symmetry was applied in a second helical refinement run, with on-the-fly refinement of local defocus and global beam aberrations, resulting in 3D reconstructions at FSC(0.143) = 3.3 Å for Atg18-PR72AA and FSC(0.143) = 3.8 Å for Atg18-WT. 3D classification and 3D variability analysis did not improve the obtained map resolutions. Soluble monomers/oligomer data (Atg18-WT and Atg18 s-loop) was picked by template matching against circular blobs (40–60 Å) in CryoSPARC live. Particle extraction was performed in 256 px/215 Å boxes to include defocus-delocalized signal. 2D classification in 200 classes was performed with increased batch sizes (200) and 100 iterations to a high-resolution limit of 4 Å. Monomeric and dimeric classes were split and subjected to ab initio 3D map generation with increased resolution ranges (12 Å to 6 Å) and increased batch sizes (300 initial, 1000 final size) with $K = 3$ classes. The monomeric class was low-pass filtered to 8 Å and subjected to 3D refinement and non-uniform local refinement[68], resulting in a map with FSC(0.143) = 4.8 Å whereas dimeric classes could not be resolved further.

Tomograms were reconstructed unbinned using IMOD/eTomo software[71] including patch tracking and weighted back projection. Subsequently, they were subjected to the PySeg pipeline[49] for particle picking and classification. Briefly, lipid membranes were segmented by TomoSegMemTV software[72], and PySeg's tracing routines were used to pick subvolumes near the segmented membranes. A total of 150,000 picked subvolumes were reconstructed in three batches and subjected to rotational 2D classification focused by a cylinder mask using the affinity propagation clustering algorithm, initially with a binning factor of 4. For double-membrane reconstructions, classes with protein signal in between the two membranes were selected and all other classes were excluded (55 classes out of 300 classes included). This way, the majority of the subvolumes were excluded as they contained single membrane classes, double-membrane classes with the second membrane outside of the focused mask as well as triple membranes with no connecting densities. For single membrane Atg18 coats only classes with a singular membrane and associated protein signal were kept and all other classes were excluded. 2D classification and class selection were repeated another time with a 2D horizontal mask suppressing the membrane signal (32 out of 516 classes included) before the extracted subtomograms were subjected to Relion v3.0 software for subtomogram averaging[73]. An ab initio 3D model was generated and refined and the PySeg tools were used for masked subtraction of the membranes (100% dampening for double membrane, 50% dampening for Atg18 coats on single membranes). Focused classification of the selected unbinned subtomograms into four 3D classes showed one dominating class for dual membranes and two dominating classes for single membrane coats that were used for 3D refinement. FDR-FSC was used to determine the nominal resolution of 26 Å of the averaged dual membrane subvolume map, 22 Å for single membrane coat class 1, and 38 for single membrane coat class 2[50]. Atg18 dimers from helical assembly were docked into the map using ChimeraX software[74,75].

### Atomic model building and refinement

PDB-ID 6KYB was docked into the reconstructed cryo-EM density of Atg18-PR72AA after map auto-sharpening in Phenix v1.20.1 in ChimeraX software. The atomic model of Atg18-PR72AA was manually mutated and slightly modified in Coot v0.9[76] before real-space refinement in Phenix was performed with default parameters[77]. Validation parameters are displayed in Table 1[78,79]. Based on the refined atomic Atg18-PR72AA model, we used the corresponding Atg18-WT model to rigidly place it into Atg18-WT tubules, Atg18 monomer as well as Atg18 subtomogram average dimer densities.

For the generation of the Atg2-Atg18 interaction model, C-terminal Atg2(Δ1–540)-Atg18 dimers were predicted using Alphafold v2.3.2[48] using the Uniref90, Mgnify_2018_12, BFD_megaclust_cu_complete_id30_c90_final_seq, uniclust30_2018_08 databases with the multimer model preset and 2022-01-07 as latest template date. All input files and the top 25 ranked output models are available at the Helmholtz data portal at https://doi.org/10.26165/JUELICH-DATA/PX1GXR. The Alphafold2 structure prediction of full-length Atg2 from UniProt P0CM30 (2022-11-01)[80] was aligned to the C-terminal prediction of the multimer and the second Atg18 molecule was docked in from the filament dimer model obtained in this study.

### Reporting summary

Further information on research design is available in the Nature Portfolio Reporting Summary linked to this article.

## Data availability

The atomic models of tubular Atg18-PR72AA and Atg18-WT were deposited in the Protein Databank (PDB-ID 8AFY and PDB-ID 8AFW), accompanied by the corresponding cryo-EM maps (EMD-15412 and EMD-15410). Single-particle maps of monomeric Atg18-WT and subtomogram average are available in the EM Databank linked to fitted PDB coordinates, respectively (EMD-15410, PDB-ID 8AFW). AlphaFold input and output files are available at the Helmholtz data portal via https://doi.org/10.26165/JUELICH-DATA/PX1GXR. Source data and supplementary datasets are provided in this paper. Source data are provided in this paper.

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

## Acknowledgements

The authors gratefully acknowledge the computing time granted by the JARA Vergabegremium and provided on the JARA Partition part of the supercomputer JURECA at Forschungszentrum Jülich[64]. The authors acknowledge Wim Hagen (EMBL Heidelberg) for assisting in setting up the tomography acquisitions of Atg18 liposome mixtures.

## Author contributions

S.A.F., D.M. and C.S. designed the research. N.G. and A.M. initially purified Atg18 and identified conditions for Atg18 tube formation. S.A.F. and R.C. purified protein performed biochemical experiments such as pelletation assay and lipid binding studies. S.A.F. and D.M. determined the presented cryo-EM structures of Atg18. D.M. and A.M.-S. computed the subtomogram average of membrane-associated Atg18. D.M. and C.S. wrote the manuscript with major input from S.A.F. and all other authors.

## Funding

## Competing interests

The authors declare no competing interests.
