## [Peer review file · Nature Communications]

REVIEWER COMMENTS

Reviewer #1 (Remarks to the Author):

Review for Mann et al “Structural plasticity of Atg18 oligomers: organization of assembled tubes and scaffolds at the isolation membrane”

Mann et al report cryoEM structures of Atg18 in tubes, Atg18 monomers and perform tomography of vesicles coated with Atg18, combined with some biochemical experiments. I think the manuscript is well written, however some of the figures are a little bit hard to interpret as they stand. While I think the paper could be published in Nature Communications, I would like to see some clarifications in terms of new figures and text before I can fully support publication. Here are the points I need some clarification on, roughly in the order in which they appear in the manuscript.

1. Figure 1

- Are there so many 2D class averages needed? Could be moved to the supplement.
- Please move the local resolution estimate to the supplement
- Important – I cannot judge at all how tightly the density and model fit each other. There is only the bottom panel of figure F that shows some beta strand separation, but not really side chain densities. The authors need to show this better. Panel F does not convince me that the resolution is 3.8 or 3.3 Å, but rather closer to 4.5 Å.

2. Atg18 – PR72AA mutant. Given the 3.8 Å wild type structure. What is this mutant adding to the paper? Should be removed or discussed only once in the main text, not throughout the first section.

3. Figure 2

- Panel B – this panel should have more copies of the atomic models and density completely removed. Because the density is not adding much to the readers’ understanding here. Just confusing.
- Panel D – what are the semi-circular grey illustrations with spokes supposed to represent? What are the green dotted lines? Explanations please. Unclear.

4. Figure 3 and accompanying text – how can the FRRG motif help interaction with membranes if it is buried?

5. Figure 4

- Panel F - This is a small protein. Why is the FSC so strange? Could you also show the negative values please after 4.8 Å? Show the full curve please.
- Panel G – again, showing the full protein and density does not allow this reviewer to judge the fit. Could you zoom on some small bits of the chain to show the fit? I want to see phenylalanine, tryptophan and other large side chains resolved in this map if you claim 4.8 Å.

6. Figure 5

- This is the weakest figure in this manuscript in my opinion. The tomography images look good, but which algorithm was used for subtomogram averaging? There is nothing mentioned in the cited reference 73.
- How were the fits validated? Were there other equivalent fits possible?
- The models do not seem to fit the density very tightly at all. There is so much empty density? How was the isosurface contour level selected?

7. Figure 6

- I am afraid this figure is not sufficiently backed up by experimental evidence and must be toned down.
- Panel D – side chain level interpretation should not be performed on an AlphaFold model fitted into a low resolution subtomogram density. While it may be correct, no experimental evidence to back it up in this manuscript.
- This figure should either be removed or toned down substantially.
- If you propose a model, then produce some further evidence to back it up. The structural biology here is very weak.

8. Suppl Figure 6 – this is an important result and could be moved to the main text.

Reviewer #2 (Remarks to the Author):

Mann and colleagues here describe the formation of higher order structures of Atg18 formed by oligomerization. The first structures described are derived from an insoluble fraction, where Atg18 (WT and PR72AA mutant) is found to assemble into helical assemblies. These helical tubes form a regular array described by "T" and "I" interfaces. These arrays are stabilized by the region of the PROPPINS which bind PI3P, the FRRG motif. The authors then go on to look at lipid binding, and determine that PIP binding disrupts the helical assemblies.

They next turn to look at the soluble fraction from their initial assays, produced by altering the salt concentration, and exploit this to obtain a monomeric structure using single particle cryo-EM. This is used to explore again lipid binding and to investigate membrane association. Addition of the soluble, monomeric protein to LUVs (95% DOPC, 5% DO-PI35P2) created membrane interfaces and deformed surfaces between which were found oligomers of Atg18. These oligomeric formations once mapped provided a basis for the modelling of Atg18 on membranes, suggesting a bridging function for Atg18s between two membranes, propose to be an isolation membrane and the ERES. Finally, the authors introduce Atg2 into the membrane-bridging model based on the established data showing Atg18 and Atg2 interact. This model is compelling although highly speculative.

The paper is well written, the data well described, and beautifully illustrated. The models are very interesting and informative in the context of the PROPPINS and recent interest in Atg2 lipid transfer function.

I have no major concerns about the experimental data. I would suggest the manuscript might be improved by the addition of data to explore the membrane interaction on LUVs. For instance, the influence of PIPs in the LUV experiments. The authors used PI35P2 to analyse the membrane binding, and it would be relevant to the cell membrane composition to test both PI, PI3P and a non-binding PIP such as PI34P2. A further interesting experiment would be to test membrane association of Atg18 lacking the hydrophobic 6CD loop. This would provide further understanding of the orientation, and membrane spacing and PIP dependence.

Reviewer #3 (Remarks to the Author):

The manuscript from Mann et al. presents an elegant investigation by cryo-EM of Atg18 oligomers in solution, as assembled tubes and reconstituted in a complex with lipid membranes. The observed structural plasticity of the Atg18 organization is suggested to have a role in positioning other components of the autophagy machinery, such as Atg2.

Up to now, there is only sparse information about the assembly of the autophagy machinery at the level of early isolation membrane and mechanisms associated with membranes and lipid transport. Thus, the manuscript results are helping to fill an important gap of knowledge and are relevant to a broad audience.

The findings and methodologies related to determining the structures of Atg18 oligomers are solid and clearly described.

My main concern is that the interpretation of the functional role of Atg18 oligomers as a structural scaffold for the binding to Atg2 and membranes needs to be supported by additional experiments to be suitable for publication. At the moment, this manuscript section is supported only by an AlphaFold multimer model of the Atg18-Atg2 complex.

WIPI4 and WIPI1 directly interact with ATG2A and have been shown by Otomo's lab to facilitate the ATG2A-mediated lipid transfer (<https://doi.org/10.7554/eLife.45777>). It will be interesting to experimentally test if Atg8 variants at the binding interfaces presented in the manuscript, such as at the level of Atg8 dimers and Atg8-Atg2, could affect Atg2 lipid transfer activity.

The section of the manuscript describing the AlphaFold multimer model of Atg2-Atg18 should be explained in more detail, including a more detailed description of the observed binding interfaces. Furthermore, AlphaFold Multimer provides confidence scores (as interface pTM score) that should be presented and discussed in the manuscript to evaluate the model accuracy.

The manuscript should include a detailed method section describing the approaches used for the modeling part including also what was used for the multiple sequence alignment used by AlphaFold. In addition, the AlphaFold Multimer inputs and outputs and associated output files should be made publicly available, such as in an OSF repository. It should also be clarified if the authors used the standalone version of the tool or they used the ColabFold notebook.

Point-by-point response (NCOMMS-22-28824).

We appreciate the overall positive feedback on our Atg18 manuscript. The three reviewers asked well-deserved questions in an effort to widen the scope of the manuscript. In this point-by-point response, we explain how we addressed the raised points by the Reviewers. In addition to the minor corrections, we added three paragraphs and Figures to illustrate the additional experiments. We also provide raw map data to the reviewers for inspection of the model fit (<https://fz-juelich.sciebo.de/s/ZExYxRZ9J2LmEb8>).

Reviewer #1 (Remarks to the Author):

Review for Mann et al “Structural plasticity of Atg18 oligomers: organization of assembled tubes and scaffolds at the isolation membrane”

Mann et al report cryoEM structures of Atg18 in tubes, Atg18 monomers and perform tomography of vesicles coated with Atg18, combined with some biochemical experiments. I think the manuscript is well written, however some of the figures are a little bit hard to interpret as they stand. While I think the paper could be published in Nature Communications, I would like to see some clarifications in terms of new figures and text before I can fully support publication. Here are the points I need some clarification on, roughly in the order in which they appear in the manuscript.

1. Figure 1

- Are there so many 2D class averages needed? Could be moved to the supplement.
- Please move the local resolution estimate to the supplement

As requested, we moved the class averages to the Supplementary Figure 1 and moved the local resolution figure from Figure 1 to the Supplement.

- Important – I cannot judge at all how tightly the density and model fit each other. There is only the bottom panel of figure F that shows some beta strand separation, but not really side chain densities. The authors need to show this better. Panel F does not convince me that the resolution is 3.8 or 3.3 Å, but rather closer to 4.5 Å.

As requested, we added side-chain densities to Figure 1 next to the separated beta-strands in the revised manuscript and provide raw map data to the reviewer for inspection of the model fit (<https://fz-juelich.sciebo.de/s/ZExYxRZ9J2LmEb8>). In β -structures, the expected features are compatible at 3.5 Å (they are different than in typical α -helical structures).

2. Atg18 – PR72AA mutant. Given the 3.8 Å wild type structure. What is this mutant adding to the paper? Should be removed or discussed only once in the main text, not throughout the first section.

Admittedly, the motivation was only referred in a very short manner. In order to clarify, we added the following statement on page 5 of the manuscript:

In an effort to improve the helical order, we imaged the previously characterized Atg18-PR72AA mutant that has lost its ability to interact with Atg2⁴⁷. The corresponding Fourier transforms...

3. Figure 2

- Panel B – this panel should have more copies of the atomic models and density completely removed. Because the density is not adding much to the readers' understanding here. Just confusing.

We removed the density from Figure 2B as requested.

- Panel D – what are the semi-circular grey illustrations with spokes supposed to represent? What are the green dotted lines? Explanations please. Unclear.

We added the following information to the legend of the Figure 2D:

Projected side-chain dimer interaction plot (DIMPLLOT) of the T (top) and I (bottom) interfaces. H-bridges and their atom distances indicated as dashes, van der Waals interactions indicated as shielded residue symbols. FRRG motif indicated as asterisks (*). Residue colors match colors in panels A,B.

4. Figure 3 and accompanying text – how can the FRRG motif help interaction with membranes if it is buried?

We argued that tube formation and membrane binding are two mutually exclusive processes in the Results on page 9:

Together, these experiments show that the FRRG motif of Atg18 takes up a stabilizing role in the formation of the helical Atg18 assembly, which is further supported by binding studies of PI3P derivatives that can prevent the assembly of Atg18 tubes and keep Atg18 soluble.

In addition, we state the context of the membrane:

The observed binding properties suggest that the helical Atg18 tubes will not form in the presence of PIP-containing membranes.

5. Figure 4

- Panel F - This is a small protein. Why is the FSC so strange? Could you also show the negative values please after 4.8 Å? Show the full curve please.

The FSC from our last submission had a strange dip because of the common way of deconvolving the effect of masks from the computation through high-resolution noise substitution (described in Chen et al. 2013). As requested, we included the negative values in the FSC and re-computed with the latest version of cryoSPARC that now shows a smooth falloff in the FSC.

- Panel G – again, showing the full protein and density does not allow this reviewer to judge the fit. Could you zoom on some small bits of the chain to show the fit? I want to see phenylalanine, tryptophan and other large side chains resolved in this map if you claim 4.8 Å.

We incorporated the suggested changes to Figure 4 and, in addition, make the raw maps and model available to the Reviewer (<https://fz-juelich.sciebo.de/s/ZExYxRZ9J2LmEb8>).

6. Figure 5 - This is the weakest figure in this manuscript in my opinion.

The tomography images look good, but which algorithm was used for subtomogram averaging? There is nothing mentioned in the cited reference 73.

This point must arise from a misunderstanding in the Methods section on page 26. At the beginning, subvolumes are extracted and averaged but no subtomogram averaging software is used for this step as it is just the addition of volumes. Therefore, we explicitly state this now:

A total of 150,000 picked subvolumes were added in three batches and subjected to rotational 2D classification focused by a cylinder mask using the affinity propagation clustering algorithm.

In fact, we had detailed all the relevant information regarding the subtomogram averaging software below:

2D classification and class selection was repeated another time with a mask suppressing the membrane signal (32 out of 516 classes included) before extracted subtomograms were subjected to Relion v3.0 software for subtomogram averaging⁷³.

- How were the fits validated? Were there other equivalent fits possible?
- The models do not seem to fit the density very tightly at all. There is so much empty density? How was the isosurface contour level selected?

We agree with the Reviewer that at this resolution an unambiguous fit on e.g. secondary structure level is not possible. However, given the limited resolution of 26 Å, the small protein size and the strong membrane signal additional structural details cannot be expected. At the same time, the dimension is already clearly visible even in the raw tomography data (Figure 6G). Interpreting the density with independently rigid body fitted Atg18 dimers taken from the high-resolution helical structure does not leave empty density and shows a good fit while reducing the degrees of freedom in the possible rigid body fitting of the Atg18 beta-propellers (Figure 6F that now contains a zoomed in view).

7. Figure 6

- I am afraid this figure is not sufficiently backed up by experimental evidence and must be toned down.
- Panel D – side chain level interpretation should not be performed on an Alphafold model fitted into a low resolution subtomogram density. While it may be correct, no experimental evidence to back it up in this manuscript.
- This figure should either be removed or toned down substantially.
- If you propose a model, then produce some further evidence to back it up. The structural biology here is very weak.

In agreement with Reviewer 2 and 3, we toned down the alphafold (AF) 2 model, removed the conclusion from the Abstract and moved the figure to the Supplementary Information (Suppl. Fig. 7). The relevant section in the Discussion was also shortened and moved to a less prominent location.

8. Suppl Figure 6 – this is an important result and could be moved to the main text.

We agree that this result is clearly backed up by the data and moved it into Figure 2E in the main Figure and added on page 7 of the Results:

Notably, four T-interfaces arrange in the characteristic lozenge pattern that was previously observed in the COPII coat formed by Sec31 tetramers (**Fig. 2E**).

Reviewer #2 (Remarks to the Author):

Mann and colleagues here describe the formation of higher order structures of Atg18 formed by oligomerization. The first structures described are derived from an insoluble

fraction, where Atg18 (WT and PR72AA mutant) is found to assemble into helical assemblies. These helical tubes form a regular array described by “T” and “I” interfaces. These arrays are stabilized by the region of the PROPPINS which bind PI3P, the FRRG motif. The authors then go on to look at lipid binding, and determine that PIP binding disrupts the helical assemblies.

They next turn to look at the soluble fraction from their initial assays, produced by altering the salt concentration, and exploit this to obtain a monomeric structure using single particle cryo-EM. This is used to explore again lipid binding and to investigate membrane association. Addition of the soluble, monomeric protein to LUVs (95% DOPC, 5% DO-PI35P2) created membrane interfaces and deformed surfaces between which were found oligomers of Atg18. These oligomeric formations once mapped provided a basis for the modelling of Atg18 on membranes, suggesting a bridging function for Atg18s between two membranes, propose to be an isolation membrane and the ERES. Finally, the authors introduce Atg2 into the membrane-bridging model based on the established data showing Atg18 and Atg2 interact.

1. This model is compelling although highly speculative.

In agreement with point 7 of Reviewer 1, we toned down the interpretation based on the AF2 predictions, removed the conclusion from the Abstract and moved the proposed model to the Supplementary Information (Suppl. Fig. 7). The relevant section in the Discussion was also shortened and moved to a less prominent location.

The paper is well written, the data well described, and beautifully illustrated. The models are very interesting and informative in the context of the PROPPINS and recent interest in Atg2 lipid transfer function.

We thank Reviewer 2 for the positive comments.

2. I have no major concerns about the experimental data. I would suggest the manuscript might be improved by the addition of data to explore the membrane interaction on LUVs. For instance, the influence of PIPs in the LUV experiments. The authors used PI35P2 to analyse the membrane binding, and it would be relevant to the cell membrane composition to test both PI, PI3P and a non-binding PIP such as PI34P2. A further interesting experiment would be to test membrane association of Atg18 lacking the hydrophobic 6CD loop. This would provide further understanding of the orientation, and membrane spacing and PIP dependence.

As suggested by the Reviewer, we added new imaging experiments with the scrambled Atg18 6CD-Loop variant (Gopaldass et al. EMBO J. 2017) and included them in a new Fig. 5 and Suppl. Fig. 4 to the revised manuscript. The results are now included on page 13 of the manuscript:

In addition to the FRRG motif, Atg18 features an amphipathic helix in its 6CD loop that was shown to insert in the membrane and ultimately involved in scission⁴⁰. In order to differentiate the two contributions of the binding properties of Atg18 to lipid membranes, we investigated samples of Atg18-WT mixed with LUVs lacking PIP(3,5)P₂ and an Atg18 6CD loop mutant (Atg18 s-loop) incapable of membrane insertion⁴⁰ mixed with LUVs containing PIP(3,5)P₂ (**Suppl. Fig. 4A**). The prepared Atg18 s-loop protein was folded correctly and formed dimers similar to Atg18-WT based on single-particle cryo-EM (**Suppl. Fig. 4B-E**). In cryo-tomograms obtained with 100% DOPC LUVs lacking PIP(3,5)P₂, we did not find an Atg18 membrane coat (**Fig. 5C**). Atg18 s-loop mixed with PI(3,5)P₂-doped LUVs formed coats on vesicles and bridged adjacent lipid membranes similarly to Atg18-WT as observed using cryo-electron tomography (**Fig. 5D**). The inter-

membrane distance in the Atg18-WT and Atg18 s-loop samples was constant over parallel membrane stretches and very similar at approx. 80 Å, respectively, whereas no consistent intermembrane spacing could be detected between undecorated 100% DOPC liposomes without Atg18 (**Fig. 5E**). Together, the binding experiments show that membrane association and inter-membrane alignment by Atg18 is primarily mediated through the FRRG-PIP interaction rather than the amphipathic helix of the 6CD loop.

We also added a more detailed analysis on the single bilayer Atg18 coats on page 14 of the revised manuscript and added a new Suppl. Fig. 6:

Moreover, we analyzed those subtomograms that did not contain two juxtaposed bilayers and processed Atg18 monolayer coats. Two resulting classes could be further averaged at a resolution of 28 Å and found that 70 % of Atg18 β-propeller particles adopted a similar 45° angle with respect to the bilayer (**Suppl. Fig. 6**).

Moreover, we discuss these results in the Discussion on page 18:

In support, when we incubated Atg18 with PIP-containing membranes, we observed the decoration of Atg18 β-propellers on lipid bilayers and the bridging of two bilayers leading to a parallel alignment of two juxtaposed bilayers (**Fig. 5**). As previously demonstrated by fluorescence microscopy, biophysical and biochemical characterizations as well as molecular dynamics simulations^{35,39,40}, Atg18 interacts with lipid membranes through the FRRG motif and the amphipathic 6CD loop in a PIP-dependent or independent manner, respectively. We found membrane binding of Atg18 in cryo-electron tomograms to be PIP(3,5)P₂-dependent whereas the amphipathic 6CD loop alone was insufficient to associate Atg18-WT to PIP-devoid membranes. Conversely, we used the well-characterized mutant Atg18 s-loop⁴⁰ to prevent 6CD-loop-dependent membrane insertion. In conclusion, Atg18 membrane association is largely PIP-dependent and the 6CD-loop alone is insufficient to anchor the protein to the membrane. Due to the demonstrated importance of the 6CD amphipathic loop⁴⁰, we putatively assign other membrane interaction roles, e.g., the ability to complete membrane fission, to this part of the protein.

Reviewer #3 (Remarks to the Author):

The manuscript from Mann et al. presents an elegant investigation by cryo-EM of Atg18 oligomers in solution, as assembled tubes and reconstituted in a complex with lipid membranes. The observed structural plasticity of the Atg18 organization is suggested to have a role in positioning other components of the autophagy machinery, such as Atg2.

Up to now, there is only sparse information about the assembly of the autophagy machinery at the level of early isolation membrane and mechanisms associated with membranes and lipid transport. Thus, the manuscript results are helping to fill an important gap of knowledge and are relevant to a broad audience.

The findings and methodologies related to determining the structures of Atg18 oligomers are solid and clearly described.

1. My main concern is that the interpretation of the functional role of Atg18 oligomers as a structural scaffold for the binding to Atg2 and membranes needs to be supported by additional experiments to be suitable for publication. At the moment, this manuscript section is supported only by an AlphaFold multimer model of the Atg18-Atg2 complex.

WIPI4 and WIPI1 directly interact with ATG2A and have been shown by Otomo's lab to facilitate the ATG2A-mediated lipid transfer (<https://doi.org/10.7554/eLife.45777>). It will be interesting to experimentally test if Atg8 variants at the binding interfaces presented in the manuscript, such as at the level of Atg8 dimers and Atg8-Atg2, could affect Atg2 lipid transfer activity.

Providing further experimental evidence as suggested by Reviewer 3 is intriguing but at the same time it is beyond the mechanistic scope of the manuscript. Given the experimental difficulties we faced over the last year when we attempted expressing Atg2 protein of a ~1500 aa in full size or fragments for structural studies, we feel such an experiment is not possible over a typical revision period. The same concern regarding the importance and validity of the AF2 model was raised by Reviewer 1, point 6 and Reviewer 2, point 1. In the first submission, the AF2 model took up a central position in the manuscript, therefore, we toned down significantly the importance of the AF2 model in the abstract and moved the figure to the Supplementary Information (Suppl. Fig. 7). The relevant section in the Discussion was also shortened and moved to a less prominent location.

2. The section of the manuscript describing the AlphaFold multimer model of Atg2-Atg18 should be explained in more detail, including a more detailed description of the observed binding interfaces. Furthermore, AlphaFold Multimer provides confidence scores (as interface pTM score) that should be presented and discussed in the manuscript to evaluate the model accuracy.

The manuscript should include a detailed method section describing the approaches used for the modeling part including also what was used for the multiple sequence alignment used by AlphaFold. In addition, the AlphaFold Multimer inputs and outputs and associated output files should be made publicly available, such as in an OSF repository. It should also be clarified if the authors used the standalone version of the tool or they used the ColabFold notebook.

We agree with the Reviewer and now provide more details on the generation of the model in the Materials and Methods section of the revised manuscript on page 27:

For the generation of the Atg2-Atg18 interaction model, C-terminal Atg2(Δ 1-540)-Atg18 dimers were predicted using Alphafold v2.3.2⁴⁸ using the Uniref90, Mgnify_2018_12, BFD_megaclust_cu_complete_id30_c90_final_seq, uniclust30_2018_08 databases with the multimer model preset and 2022-01-07 as latest template date. All input files and the top 25 ranked output models are available at the Helmholtz data portal at <https://doi.org/10.26165/JUELICH-DATA/PX1GXR>. The Alphafold2 structure prediction of full-length Atg2 from UniProt P0CM30 (2022-11-01)⁸⁰ was aligned to the C-terminal prediction of the multimer and the second Atg18 molecule was docked in from the filament dimer model obtained in this study.

REVIEWERS' COMMENTS

Reviewer #1 (Remarks to the Author):

The authors have provided appropriate responses to my comments and I support publication of this article.

Reviewer #2 (Remarks to the Author):

The authors have nicely addressed my comments.